# Designing a hybrid dimension reduction for improving the performance of Amharic news document classification

**Demeke Endalie** [1] *, **Tesfa Tegegne** [2]

**1** Factuality of computing and Informatics, Jimma institute of technology, Jimma, Ethiopia, **2** Factuality of computing, Bahir Dar Institute of Technology, Bahir Dar, Ethiopia

* demeke.endalie@ju.edu.et

**Data Availability Statement:** All relevant data are uploaded to GitHub and accessible via the following URL: https://github.com/demekeendalie/feature-selection.

## Abstract

The volume of Amharic digital documents has grown rapidly in recent years. As a result, automatic document categorization is highly essential. In this paper, we present a novel dimension reduction approach for improving classification accuracy by combining feature selection and feature extraction. The new dimension reduction method utilizes Information Gain (IG), Chi-square test (CHI), and Document Frequency (DF) to select important features and Principal Component Analysis (PCA) to refine the features that have been selected. We evaluate the proposed dimension reduction method with a dataset containing 9 news categories. Our experimental results verified that the proposed dimension reduction method outperforms other methods. Classification accuracy with the new dimension reduction is 92.60%, which is 13.48%, 16.51% and 10.19% higher than with IG, CHI, and DF respectively. Further work is required since classification accuracy still decreases as we reduce the feature size to save computational time.

## Introduction

Amharic is a Semitic language and the working language of Ethiopia (ኢትዮጵያ) [1]. Amharic has more than 25 million speakers in Ethiopia, but there are also millions of Amharic speakers in a number of other countries, particularly Eritrea, Canada, USA and Sweden [2]. Nowadays, the volume of the Amharic digital document has increased rapidly. Due to this, an advancement in information comes up with the issue of generating, storing, retrieving, printing and publishing of individual documents faster and simpler than ever before [3]. The act of labeling or tagging documents based on their content is known as document classification. It is one of the natural language processing (NLP) application which plays a great role for flexible and dynamic management of information [4].

One major challenge in document classification using a machine learning model is dealing with dimensionality. This is due to the fact that the majority of existing machine learning algorithms are not designed to work with a large number of features [5]. The dimensionality of a dataset is defined as the number of input variables or features. The dimension of the input features influences the model's overall performance. Dealing with a high-dimensional feature

**Funding:** The author(s) received no specific funding for this work.

**Competing interests:** The authors have declared that no competing interests exist.

space, analyzing and visualizing patterns becomes difficult while training the learning model. As a result, reducing dimensionality is a critical task in the classification process. It is used to save computational and storage resources while processing and analyzing features, as well as to avoid underfitting and overfitting.

Text documents can contain hundreds or thousands unique terms. If we used all of the terms in the classification, we may get a poor result because some of the terms are useless for classification. In document classification, two fundamental dimension reduction techniques are introduced: feature selection and feature extraction. The process of removing irrelevant, redundant, or noisy features from a large set of original features in order to select a small sub-set of relevant features is known as feature selection. Feature selection is divided into three main categories: filter, wrapper and embedded [6]. In filter approach, features are selected based on their relevance with feature scoring metrics such as DF, IG, CHI, and term strength. The wrapper approach, on the other hand, selects a feature subset by using the accuracy of the classifier as a guiding criterion [7]. The filter approach ignores feature interaction. However, it is the most widely used feature selection technique because it is fast, scalable with high dimensional data, has better generality, and is less computationally complex than the wrapper. As a result, we employ the filter approach to select important terms for classification.

Feature extraction is a dimensionality reduction process that reduces an initial set of raw data to more manageable groups for processing. Feature extraction refers to methods that select or combine variables to form features, thereby reducing the amount of data that must be processed while accurately and completely describing the original data set. The most commonly used feature extraction techniques are PCA, Latent Semantic Indexing (LSI), Independent Component Analysis (ICA), Linear Discriminant Analysis (LDA), multi-dimensional scaling, and Partial Least Squares (PLS) [8].

Many dimension reduction methods were discussed in order to improve the classification accuracy of English and Arabic text [9–11]. However, there are only a few dimension reduction works done for Amharic document classification [12–14]. Their primary emphasis was on the selection of the classifier algorithm, rather than the number and quality of features used by the learning model. Feature selection methods such as IG, CHI, and DF can be used to extract the most relevant features from a dataset, and feature extraction methods such as PCA can be used to convert a feature matrix to a low-dimensional matrix with no information loss.

As a result, the goal of this paper is to present a hybrid dimensionality reduction method for Amharic document classification in order to improve classification accuracy while reducing the number of input features. The proposed dimension reduction method includes feature selection methods such as IG, CHI, and DF, a union operator to combine highly-ranked features, and an intersection operator to join least-ranked features selected by IG, CHI, and DF, followed by PCA on the document-term matrix generated by the unified feature set to reduce the number of features before they are passed to the learning algorithm. The main contribution of this paper are as follows:

1. Presenting a detailed review of the literature on dimensionality reduction techniques.

2. Propose a dimension reduction system that incorporates IG, CHI, and DF with PCA, as well as a new union-intersection feature merging strategy that utilizes union and intersection.

3. Examine the performance of various dimension reduction methods (IG, CHI, DF, and PCA) on machine learning algorithms in terms of accuracy, precision, recall, and F-measure.

4. Prove that the proposed hybrid dimension reduction method do not degrade the performance of machine learning algorithm in a great extent.

The remainder of the paper is structured as follows: Section II is a literature review. Section III describes the dimensionality reduction techniques and methodology used in this work. The results of the experiments are presented and discussed in Section IV. Section V focuses on the conclusion and future work.

## Materials and methods

### Related works

The accuracy of the classifier algorithms used in Amharic news document classification is affected by dimension reduction method used. A number of research studies have attempted to use different dimension reduction techniques to overcome the problem of curse of dimensionality.

Worku [14] conducted a research on automatic Amharic news classification using Artificial Neural Network (ANN). The author tries to see the potential application of learning vector quantization over Amharic document classification. The author uses single dimension reduction technique i.e. DF and manual selection of term which are key word for one class and do not catch up with the defined DF threshold value. In the previous studies of Amharic text classification, only single dimension reduction technique was used to reduce the feature space, which may increase the computational cost, memory storage and underfitting or overfitting.

The authors [15] proposed a more accurate ensemble classification model for detecting fake news. The proposed model extracts important features from fake news datasets and then classifies them using an ensemble model composed of three popular machine learning models: Decision Tree, Random Forest, and Extra Tree Classifier. However, ensemble classifiers require an inordinate amount of time to train the ensemble classifier model.

The work in [9] proposed a dimension reduction by integrating feature selection with feature extraction. They selected features using DF and Term Variance (TV), and extracted using PCA. They used global thresholding while applying DF and VT. Their method shows better performance. Nevertheless, features selected by global thresholding results biased towards to the most frequently occurring news category [16].

The authors [17] investigate big data dimensionality reduction. They compare and contrast the two feature extraction methods PCA and Linear Discriminant Analysis (LDA) using different machine learning algorithms on Cardiotocography (CTG) dataset. Their results demonstrated that using PCA-SVM works well. Still, using PCA for text classification is computationally expensive [10].

The work of [11] proposed a feature selection method for Arabic text classification that uses an improved Chi-square to improve classification performance. They contrasted their improved feature selection method with three other feature selection metrics: mutual information, IG, and CHI. The authors put their findings to the test on a dataset of 5070 Arabic documents that were classified into six distinct classes using an SVM classifier. The proposed method improves the performance of an Arabic text classification model. The best f-measures obtained from their model are 90.50% when the number of features is 900. However, 900 is still a large number, and training the classifier takes time.

The authors of [18] proposed a hybrid Latent Semantics Indexing (LSI)-based Feature Selection Approach for Urdu Text Classification. To classify Urdu text, they used SVM classifier. The authors combine the CHI, IG, and Gain Ratio (GR) feature selection methods with LSI. They test their proposed method with a dataset of 29,931 news articles divided into 16

**Table 1. Summary of papers.**

| Papers | Methods | Finding | Limitations |
|---|---|---|---|
| [9] | DF+TV+PCA and K-means | Introduced a hybrid feature selection method that combines the advantages of one method while mitigating the drawbacks of the other. The new dimension reduction improves the clustering accuracy. | They used global thresholding, which results in a bias toward news categories with more instances in the corpus. |
| [11] | Improved CHI and SVM | The authors put their findings to the test on a dataset of 5070 Arabic documents that were classified into six distinct classes using an SVM classifier. The best f-measures obtained from their model are 90.50% with 900 features. | Their dimension reduction results in a greater number of features, which requires a longer training time. |
| [14] | DF and Learning Vector Quantization | The author tries to see the potential application of learning vector quantization over Amharic document classification using two basic term weighting algorithm: TF and TF-IDF. | They used DF and manual selection of some key terms, resulting in a non-automatic system. |
| [15] | Ensemble classifiers which consists of Decision Tree, Random Forest, and Extra Tree Classifier | Authors achieved training and testing accuracy of 99.8% and 44.15%, respectively, on the ISOT dataset and 100% testing and training accuracy on Liar dataset. | Improving classification accuracy with an ensemble classifier requires a longer training period. |
| [17] | PCA and LDA with SVM, Random Forest, Decision tree | Dimensionality reduction techniques have a negative impact on the performance of ML algorithms when the dataset size is too small. When the size and dimensions of the dataset are significant PCA performs better than pure classifiers without dimensionality reduction. Using PCA for better results in terms of specificity, sensitivity and accuracy metrics | Applying feature extraction techniques such as PCA and LDA to a corpus with thousands of inputs takes a large amount of time. |
| [18] | IG+CHI+GR and LSI with SVM | The proposed method produced better results in terms of dimensionality reduction. The classification of Urdu text was performed using well-known SVM classifiers, and the results were found to be comparatively satisfactory, with an accuracy of 62.57%. | They did not test the consistency of their proposed dimension reduction model on different classifiers. |

categories in Urdu. They conclude that their proposed method produces a better classification with promising accuracy. They did not test their proposed method with various classifiers.

When we reviewed previous works on Amharic document classification and dimension reduction, we noticed that single dimension reduction was used to minimize the input features. A single dimension reduction cannot take multiple aspects of the same document at the same time. In this study, we explore a dimension reduction scheme which consists of FS-FS-FS-FE dimension reduction scheme that reduces the computational time without affecting the classification accuracy. The summary of related works and their limitations is shown in Table 1 below.

## The proposed hybrid dimension reduction approach

The processing of the proposed dimension reduction method start with collection of documents. Here, we use news document collected form Ethiopian News Agency (ENA). The proposed architecture of automatic Amharic document classifier used by this study consists of five basic components as shown in Fig 1. These components are the pre- processing, document representation, dimensionality reduction, feature weighting and lastly the classifier module. The pre-processing module has tokenizer, normalization, stop-word removal, and stemmer subcomponents. The dimension reduction component has feature selection and feature extraction modules. The weighting module assigns weight to those terms selected by the feature selection stage of dimension reduction. MLP is used for classification of documents. The algorithm1 below describes the functionality of the proposed system.

```
Algorithm1: the proposed dimension reduction method
Input: Raw document D
Begin
```

```
    1. Pre-process D (normalization, tokenization, stop word removal
and stemming)
    2. Make a list of unique words in D.
    3. Determine the IG, CHI, and DF value of each word.
    4. Define the IG, CHI, and DF thresholding values.
    5. Select terms with IG, CHI, and DF values greater than or equal
to the threshold value.
    6. Sort word lists in ascending order by IG, CHI, and DF scoring
values.
    7. Divide the above word lists into two parts using 75-25 ratio.
    8. Merge the first parts by intersection operator and union opera-
tor for the second parts.
    9. Combine the result of step 8 with the union operator.
    10. Create a document-term matrix with TF-IDF weighting.
    11. Implement PCA on the matrix at step 10.
    12. Divide the data by 80-20 for training and testing.
    13. Train with a training set.
    14. Evaluate with testing set.
  End
```

The overall architecture of the model is shown in Fig 1 below.

**Data pre-processing module.**   Previous works used Ethiopic Representation in ASCII (SERA) to convert Amharic character ("ፊደል") to Latin script; however, we have used Amharic characters in every step of document processing. This preserves the integrity of the original document. Subcomponents under pre-processing module are presented as follows.

**Normalization**: Amharic is one of the morphological rich Semitic languages [19]. In Amharic, there are different characters, which has the same sound and meaning. These characters include ሀ፣ሃ፣ኀ፣ኌ፣ሐ፣ሓ, ሰ፣ሠ, አ፣ኣ፣ዐ፣ዓ, ጸ፣ፀ. Writing name of an object or concept with different alphabets/characters results in increase the size of the feature. For example, the entity "Sun" can be written in different words like ጸሃይ፣ ጸሐይ፣ ጸሀይ፣ጸሓይ፣ጸኀይ፣ጸኃይ፣ፀሃይ፣ ፀሐይ፣ፀሀይ፣ፀሓይ፣ፀኀይ፣ፀኃይ but they did not have semantic difference. To reduce the feature space

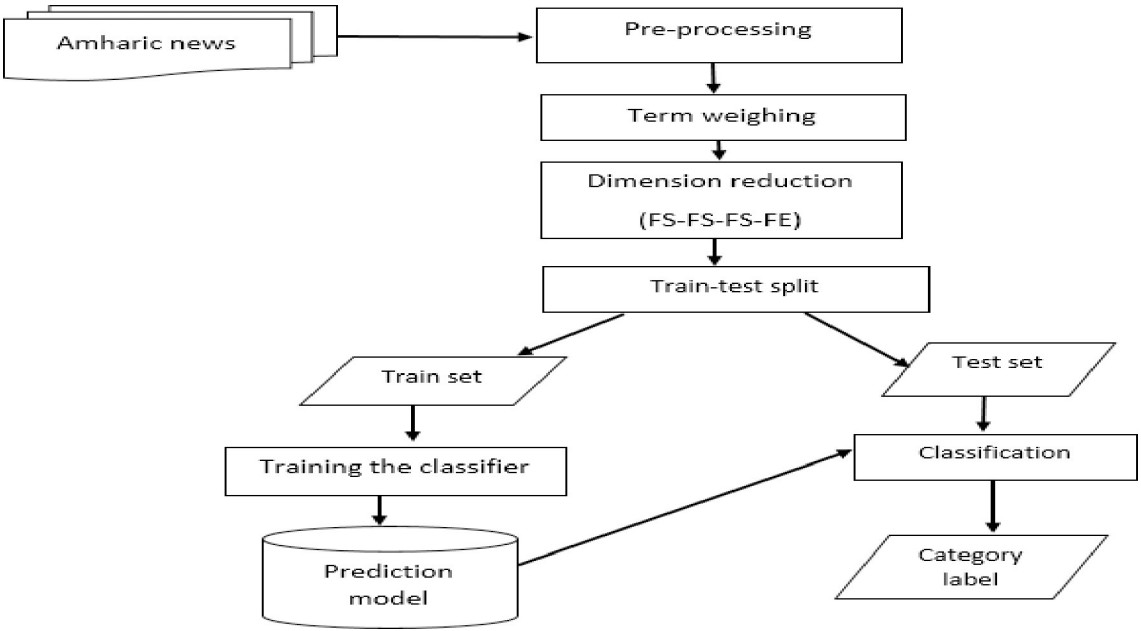

**Fig 1. Architecture of Amharic document classification.**

in this study, we prepare a list of characters that have the same sound with their common form used by this study.

**Tokenization**: As noted by Nitin [4], tokenization is the process of identification of all the individual words of a document. In Amharic, punctuation marks and spaces are used to indicate the beginning and ending of tokens. Terms in Amharic documents are identified by Amharic word separators or punctuation marks such as single space, netela serez (፣), hullet netb (፥), dirb serez (፤), arat netb (።), carriage return, line feed, tab, etc. [20] For document tokenization first we replace all Amharic word separators with single space. Then the sequence of character followed by space, four dots and newline is taken as a word or a token.

**Stop-word removal:** In this study, stop-words are removed in two ways. One is removing a list of non-content bearing terms from the whole feature set and the other is omitting features that did not fulfil the evaluation function on feature selection stage. Tokens that contain numbers or digits are also considered as stop-words in our case. The list of stop-words used in this study is collected from Amharic document processing works done previously by Eyob [21].

**Stemming:** This is a process of reducing morphological variants of a word into its root form. Amharic is morphologically rich language that contains prefixes such as የ፣በ፣ስለ፣እንደ፣ከ፣ወደ, infixes such as ላ፣ጃ፣ቃ, suffixes such as ኦች፣ኦችም፣ን፣ንም፣ናም፣ኡ and derivatives. In this study, we have used HornMorpho 3.1 stemmer [19]. HornMorpho is a Python program developed by Michael Gasser that analyzes Amharic, Oromo and Tigrigna words to their constituent morphemes (meaningful parts) and generates words, given a root or stem and a representation of the word's grammatical structure.

**Dimension reduction module.** After performing all the pre-processing tasks, we need to reduce the number of features to lower computational complexity and higher classification accuracy. Different hybrid dimension reduction model were found in the literature [9,22]. In this study, we apply three different feature selection methods (DF, IG and CHI) to get three different feature sets which contains terms that have different characteristics. This helps to catch up terms omitted by one feature selection method with the other. Then apply local thresholding on IG, DF and CHI scoring metrics in each news category. In the following section we describe each feature selection method, hybrid FS model and feature merging strategy.

*Information Gain.* Information gain is used to know term goodness in machine learning. It measures the bit of information obtained for categories by knowing the presence or absence of a given term in a document. In other word information gain measure for finding the worthiness of features for classification and the IG of a term t can be calculated as follows [23].

$$IG(t) = -\sum_{i=1}^{m} P(Ci)logP(Ci) + P(t)\sum_{i=1}^{m} P(Ci|t)logP(Ci|t) + P(\neg t)\sum_{i=1}^{m} P(Ci|\neg t)logP(Ci|\neg t) \quad (1)$$

Where m is the number of categories, P(Ci) is the probability of the i[th] category, P(t) and P(¬t) are the probabilities of presence and absence of term t, and P(Ci|t) and P(Ci|t) the probability of Ci with the presence or absence of term t respectively

*Chi-square test.* The CHI test is a statistical technique for determining the degree of independence between two events. The Chi-square test is used to determine whether a particular feature and a particular class are independent [24]. If the two are interdependent, we can use that feature to predict the class's occurrence. The higher the CHI score, the more likely the feature is related to the class. Mathematically, CHI (Ci, tj) can be calculated as follows:

$$CHI(Ci, tj) = \frac{N(AD - CB)^2}{(A + C)(B + D)(A + B)(C + D)} \quad (2)$$

where N is the total number of documents in the collection, A is the number of documents in class Ci that contain the term tj, B is the number of documents in other classes that contain the

term tj, and C is the number of documents in class Ci that do not contain the term tj; D is number of documents that do not contain the term tj in other classes.

*Document frequency*. DF thresholding is simple with low cost of computation to select features. The basic idea behind DF is terms, which found in lesser number of documents are not important for the classification process. For each term, its DF value is the number of documents containing that specific term. After that take all the features having DF value greater that the predefined threshold.

$$DF = \sum_{i=1}^{m}(A_i) \tag{3}$$

Where Ai = document i where the word is present, m = number of documents, and i is an integer ranging from 1 to m.

*Hybrid FS model and feature merging*. The feature subsets generated by each method are combined to form a single set that is used by the next classification process. The following are the steps in the hybrid FS model. The feature selection method and feature merging strategy used in this study are described in the algorithm2 below.

```
Algorithm2: Hybrid feature selection and feature merging
Input: pre-processed feature set
Begin:
    1. Implement IG, CHI, and DF and keep all features with scores
higher than the threshold value.
    2. Repeat step 1 for each feature selection method to obtain IG-
set, CHI-set, and DF-set.
    3. Sort IG-set, CHI-set and DF-set in ascending order.
    4. Divide the IG-set, CHI-set, and DF-set by a 75-25 ratio.
    5. Use intersection operator to combine the first parts and union
operator to combine second parts created in step 4.
    6. Combine the results obtained in step 5.
End
```

Fig 2 shows a graphical description of the new dimension reduction module (IG+CHI+DF +PCA) used in this study.

**Feature weighting.** To classify textual documents, a vector representation model is used to map them to vectors [9]. The importance of terms in the classification is measured using Term Frequency by Inverse Document Frequency (TF-IDF). Since it took the advantage of Term Frequency (TF) and Inverse Document Frequency (IDF) [25]. Mathematically, this method of term weighting can be formulated as follows.

$$TF - IDF(t,d) = TF_{t,d} * \log N/DF_t \tag{4}$$

Where $TF_{t,d}$ denotes the term frequency of term t in document d, N denotes the total number of documents in the corpus, and $DF_t$ denotes the document frequency of term t in the collection.

*Document-term matrix*. After pre-processing and selecting the relevant terms from each category using the above feature selection methods, we represented each document as a matrix. The document-term matrix describes the frequency with which terms appear in documents. The generated document-term matrix is presented in an excel sheet. The document-term matrix is made up of columns of terms and rows of documents. It has N by R size, where N is number of documents in the collection and R is number of terms. The TF-IDF of the given term in the respective document is stored in the matrix. A new column is added to the matrix to store the news category label. Table 1 shows a sample snapshot of document-term matrices for the first news category's (Economy) news documents. In this study, we used numbers ranging from 0 to 8 to represent the category labels of the following news categories:

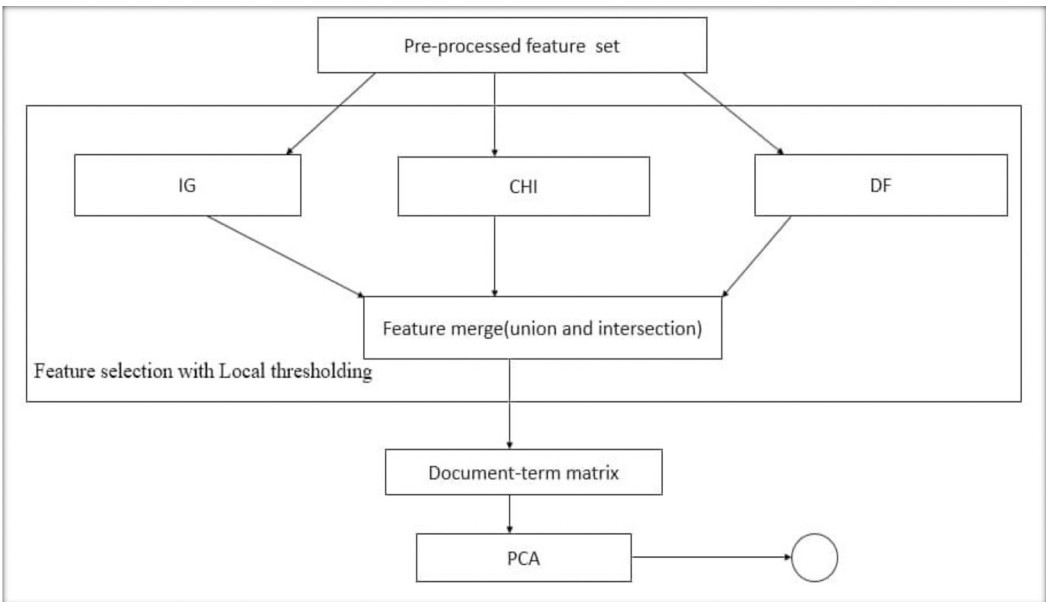

**Fig 2. Pictorial description of the proposed dimension reduction scheme.**

*Economy*, *Education*, *Sport*, *Culture*, *Accident*, *Environmental Protection*, *Diplomatic Relation*, *Justice*, and *Agriculture*. Table 2 displays a sample document-term matrix for five documents from the Economy category.

*Feature extraction.* We used PCA to reduce the input dimension or number of features after obtaining the document-term matrix. PCA allows us to reduce the dimensionality of the feature space without compromising classification performance [9]. Before reducing features, we must determine the optimal number of principal components to be used in PCA. We count the number of components that handle 97% of the total variance. We chose a variance ratio of 97% because variance ratios between 95% and 99% are considered best practice by the PCA [26].

## Results and discussion

The experiment is carried out on Amharic news document classification obtained from ENA using Python 3. A personal laptop with a corei7 processor, Windows 10 operating system, and 16GB RAM is used for this experiment.

### Dataset description

The dataset for this study is collected from the ENA from year 2018–2019 (GC). We chose ENA as a dataset source for testing because it is easily accessible on the internet and simple to collect clear Amharic news documents. ENA documents come in a variety of sizes. Some news

**Table 2. Sample document-term matrix.**

| Documents | Representative terms | | | | | | | Label |
|---|---|---|---|---|---|---|---|---|
| | ንግድ/Trade | ስራ/Work | ግብር/Tax | መንገድ/Way | ልማት/Growth | ገበያ/Market | ምርት/Product | |
| Doc1 | 0.01557 | 0.03771 | 0.00122 | 0.00435 | 0 | 0 | 0.0254 | 0 |
| Doc2 | 0 | 0.02679 | 0.01107 | 0.01152 | 0.05436 | 0 | 0 | 0 |
| Doc3 | 0 | 0.0494 | 0.04779 | | 0 | 0.09827 | 0 | 0 |
| Doc4 | 0 | 0.02938 | 0 | | 0 | 0.23413 | 0.133 | 0 |
| Doc5 | 0.14333 | 0.04973 | 0 | | 0.01251 | 0.0124 | 0.25 | 0 |

**Table 3. News categories label and number of news document in each news category.**

| labels | Category | No. document | No. training document | No. testing document |
|---|---|---|---|---|
| 0 | Economy | 217 | 173 | 44 |
| 1 | Education | 209 | 175 | 34 |
| 2 | Sport | 220 | 183 | 37 |
| 3 | Culture | 205 | 162 | 43 |
| 4 | Accident | 207 | 161 | 46 |
| 5 | Environmental protection | 210 | 165 | 45 |
| 6 | Diplomatic relation | 205 | 162 | 43 |
| 7 | Justice | 145 | 117 | 28 |
| 8 | Agriculture | 205 | 160 | 45 |
| **Total** | | **1,823** | **1,458** | **365** |

documents may contain as few as four sentences, while others may contain as many as twenty sentences. We took all the news from 2018 to 2019 because of older news are removed from their website as new news are posted. Each document file is saved in separate file name within the corresponding category's directory: i.e., all documents in the dataset are single labeled. We obtained our data from the link: https://www.ena.et. The statistical summary of the dataset used in this study is presented in Table 3 below.

## Performance measure

The classification algorithm's performance with our dataset was evaluated using the accuracy, precision (P), recall (R), and F-measure (F) [27].

**Accuracy**: This is the most common metric for classifier efficiency, and it can be calculated as follows:

$$Accuracy = \frac{TP + TN}{TP + TN + FP + FN} \tag{5}$$

**Precision**: used to determine the correctness of a classifier's result and can be determined as follows:

$$Precision = \frac{TP}{TP + FP} \tag{6}$$

**Recall**: the completeness of the classifier results is measured by recall. The following equation is used to calculate it:

$$Recall = \frac{TP}{TP + FN} \tag{7}$$

**F-measures**: it is the harmonic mean of precision and recall, and it can be calculated as follows:

$$F - measure = \frac{TP + TN}{TP + TN + FP + FN} \tag{8}$$

**Table 4. MLP classifier confusion matrix.**

| Label | Precision | Recall | F1-score | Support |
|---|---|---|---|---|
| 0 | 1.00 | 1.00 | 1.00 | 44 |
| 1 | 1.00 | 1.00 | 1.00 | 34 |
| 2 | 1.00 | 0.97 | 0.99 | 37 |
| 3 | 0.91 | 0.93 | 0.92 | 43 |
| 4 | 0.93 | 0.89 | 0.91 | 46 |
| 5 | 0.94 | 0.69 | 0.79 | 45 |
| 6 | 0.82 | 0.98 | 0.89 | 43 |
| 7 | 0.78 | 1.00 | 0.88 | 28 |
| 8 | 0.98 | 0.93 | 0.92 | 45 |

## Performance evaluation of classifiers with the new dimension reduction

This section discusses the results of the experiments. The dataset with the new dimension reduction is tested using four machine learning techniques: SVM, MLP, and Decision Tree (DT). The confusion matrices over the dataset with the new dimensionality reduction are shown in Tables 4–6 below.

Fig 3 shows the accuracy of the above-mentioned algorithms on the dataset. As Fig 3 shows that MLP outperforms SVM and DT classifiers. The classification accuracy of MLP, SVM and DT is 90.68%, 87.67% and 75.34% respectively.

In the next phase, we used MLP classifier to compare the new dimension reduction performance to that of traditional dimension reduction methods such as IG, CHI, DF, and pure PCA. IG, CHI, DF, pure PCA, and the new dimension reduction produce 540, 746, 393, 1635, and 194 features, respective.

The new dimension reduction method outperforms other traditional methods in terms of accuracy, precision, recall, and F-measure (as shown in Fig 4). The new dimension reduction method chooses important terms by employing three different feature scoring metrics to obtain different aspects of the same document, as well as PCA to reduce the number of features without risking classification accuracy.

According to Table 7, the performance of the classifier algorithms with the new dimension redaction outperforms others with smaller numbers. The number of features produced by the new dimension reduction is less than half of the number produced by DF (smaller from others).

To summarize the preceding findings and discussion, using multiple dimension reductions at the same time improves classification accuracy. At the same time, it reduces the number of

**Table 5. SVM classifier confusion matrix.**

| Label | Precision | Recall | F1-score | Support |
|---|---|---|---|---|
| 0 | 0.74 | 0.84 | 0.79 | 44 |
| 1 | 0.97 | 0.85 | 0.91 | 34 |
| 2 | 1.00 | 0.97 | 0.99 | 37 |
| 3 | 0.88 | 0.81 | 0.84 | 43 |
| 4 | 0.91 | 0.91 | 0.91 | 46 |
| 5 | 0.84 | 0.84 | 0.84 | 45 |
| 6 | 0.88 | 0.98 | 0.92 | 43 |
| 7 | 0.85 | 0.82 | 0.84 | 28 |
| 8 | 0.88 | 0.84 | 0.86 | 45 |

**Table 6. DT classifier confusion matrix.**

| Label | Precision | Recall | F1-score | Support |
|---|---|---|---|---|
| 0 | 0.74 | 0.80 | 0.77 | 44 |
| 1 | 0.86 | 0.91 | 0.89 | 34 |
| 2 | 0.85 | 0.89 | 0.87 | 37 |
| 3 | 0.65 | 0.56 | 0.60 | 43 |
| 4 | 0.84 | 0.91 | 0.87 | 46 |
| 5 | 0.51 | 0.53 | 0.53 | 45 |
| 6 | 0.84 | 0.84 | 0.84 | 43 |
| 7 | 0.61 | 0.68 | 0.64 | 28 |
| 8 | 0.84 | 0.69 | 0.76 | 45 |

features; for example, IG+CHI+DF produces 405 features with an accuracy of 88.71%, whereas IG+CHI+DF+PCA produces 194 features with an accuracy of 92.60%. This is because combining different dimension reduction techniques improves classifier performance by balancing the drawbacks of one technique with the strengths of the other.

## Conclusion

This paper describes an Amharic document classification dimension reduction system that combines three feature selection methods (IG, CHI, and DF) and a feature extraction method (PCA). The new merging system achieves the desired result by applying thresholding for the three feature selection method on a per-news-category basis. To find the most representative terms, the system uses IG, CHI, and DF relevance score of the term. The experimental analysis is carried out on 1823 Amharic news documents in order to categorize them into 9 categories. The new dimension reduction method reduces the feature set to 194, which is half the size of the smallest feature subset produced by DF (393 features). The new dimension reduction method reduces the time and space required to train the classifier by reducing the number of features produced by IG, CHI, and DF by 64.07%, 74%, and 50.63%, respectively. As a result,

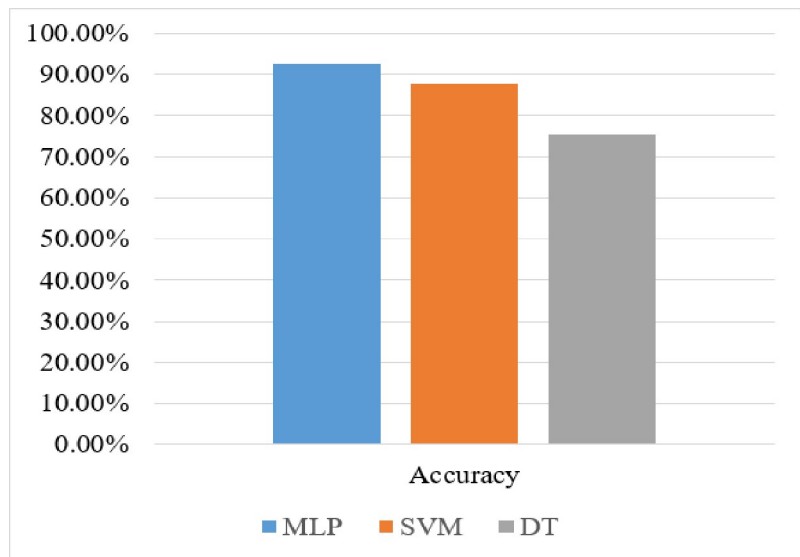

**Fig 3. Performance evaluation of classifiers with the new dimension reduction method.**

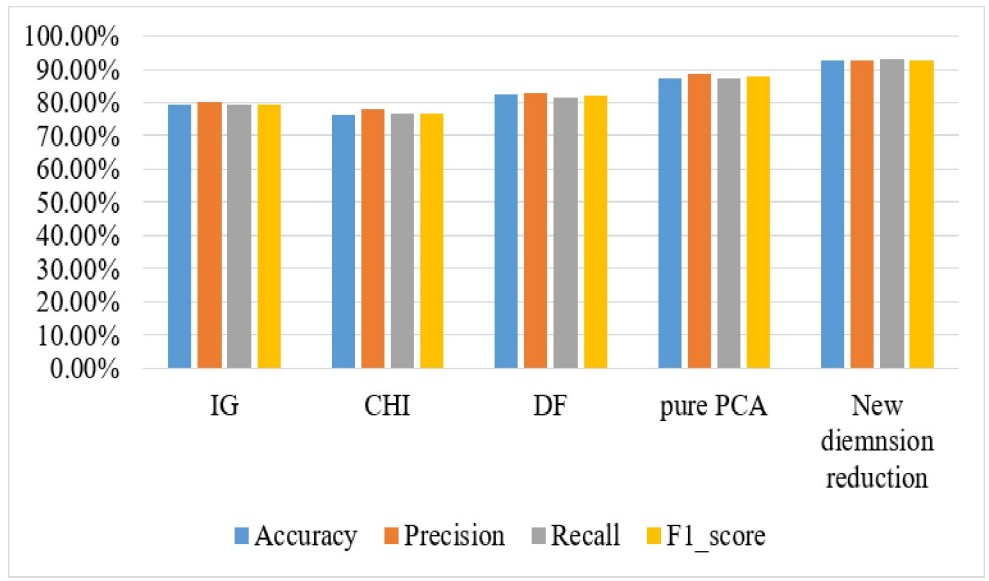

**Fig 4. Comparison of traditional dimension reduction with the new dimension reduction.**

**Table 7. Summary of dimension reduction on our dataset.**

| Classifier | Dimension reduction methods | Accuracy | Number of features |
|---|---|---|---|
| MLP | new dimension reduction (IG+CHI+DF+PCA) | 92.60% | 194 |
| | IG | 79.12% | 540 |
| | CHI | 76.09% | 746 |
| | DF | 82.41% | 394 |
| | Pure PCA | 87.45% | 1635 |
| | IG+CHI+DF | 88.71% | 405 |
| | Without dimension reduction | 89.60% | 15,178 |

the proposed dimension reduction method is suitable for use in a variety of applications requiring Amharic document classification, such as automatic document organization, topic extraction, and information retrieval. However, some enhancements to new dimension reduction could be taken to reduce the loss in classification accuracy as the number of features and categories increases. Semantic dependency among of features will be investigated further in our future work.

## Acknowledgments

This study was carried out by two academic staffs from Jimma University and Bahir Dar University in Ethiopia. The authors would like to thank the institutes for their assistance with various resources.

## Author Contributions

**Conceptualization:** Demeke Endalie.

**Data curation:** Demeke Endalie.

**Formal analysis:** Demeke Endalie.

**Investigation:** Demeke Endalie.

**Methodology:** Demeke Endalie.

**Supervision:** Tesfa Tegegne.

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
