## [Decision Letter · Decision Letter 0]

12 Apr 2021

PONE-D-21-09681

Designing a Hybrid Dimension Reduction for Improving the Performance of Amharic News Document Classification

PLOS ONE

Dear Dr. Endalie,

Thank you for submitting your manuscript to PLOS ONE. After careful consideration, we feel that it has merit but does not fully meet PLOS ONE’s publication criteria as it currently stands. Therefore, we invite you to submit a revised version of the manuscript that addresses the points raised during the review process.

Based on the comments from the reviewers and my own observation, I recommend major revisions for this paper.

We look forward to receiving your revised manuscript.

Kind regards,

Thippa Reddy Gadekallu

Academic Editor

PLOS ONE

Journal Requirements:

Reviewers' comments:

Reviewer's Responses to Questions

**Comments to the Author**

1. Is the manuscript technically sound, and do the data support the conclusions?

Reviewer #1: Yes

Reviewer #2: Yes

Reviewer #3: Yes

Reviewer #4: Yes

2. Has the statistical analysis been performed appropriately and rigorously? 

Reviewer #1: No

Reviewer #2: Yes

Reviewer #3: Yes

Reviewer #4: Yes

3. Have the authors made all data underlying the findings in their manuscript fully available?

Reviewer #1: Yes

Reviewer #2: Yes

Reviewer #3: Yes

Reviewer #4: Yes

4. Is the manuscript presented in an intelligible fashion and written in standard English?

Reviewer #1: No

Reviewer #2: Yes

Reviewer #3: Yes

Reviewer #4: Yes

5. Review Comments to the Author

Reviewer #1: Authors have presented a dimension reduction approach by integrating feature selection with feature extraction.

Even though at some places the paper is well-written, it misses out at many other sections.

Contributions and novelty are major aspects of a research work to be accepted; I find those lacking.

I strongly encourage the authors to majorly revise the manuscript before resubmission and come up with strong literature review, presentation of the algorithms, mathematical model and rigorous analysis of the results.

For now, I cannot recommend the manuscript for revisions.

Reviewer #2: - Abstract needs to be reduced. It should be between 150-200 words

- 1. Please improve the abstract. It should highlight the background as well and add some factual results that shows how much results have your approach achieved.

2. Introduction is too long. Please reduce it. It should be specific.

3. Improve the contribution list at the end of the introduction and organization paragraph.

4. Result section is written badly. Explain results extensively and clearly explain the flow of results.

5. Explain the dataset in detail and add refereces. what is the motiation behind selecting this dataset?

6. Abbreviations and Acronyms should be added only first time. Then only acronyms should be used in the entire paper.

7. Few latest references can be added in the related work.

8. Problem statement must be in the introduction.

9. Grammar check is required.

- Authors should add the most recent reference:

1)Cross corpus multi-lingual speech emotion recognition using ensemble learning, Complex & Intelligent Systems

2) BCD-WERT: a novel approach for breast cancer detection using whale optimization based efficient features and extremely randomized tree algorithm, Peerj sciences

Reviewer #3: The authors have presented a a new dimension reduction approach by integrating feature selection with feature extraction. This paper is suitable for publication but it needs minor revision.

Below are my comments:

• The contributions of the authors are not clear. They have mentioned in first contribution.

• Several paragraphs contain trivial information and should be dropped.

• Each section should have a summary table. If contents are too much, then add summary tables for the subsection.

• Each section should present new information and perspective to enlighten the readers.

• Paper contributions are presented without pitching problems in recent studies. Add one paragraph before it to highlight issues in recent studies and at the end how this paper overcome those shortcomings.

• Improve the presentation and resolution of table 1. It’s a very informative figure.

• Figures should be improved as its a very important figure that shows the reduction scheme

• writing is good, need to check the typo errors.

• paper is well-formatted, plz check the formatting of the reference

• equations should be explained well and checked again

• I found some English mistakes please check them.

• There is relevant literature missing. Please cite articles below but not limited to:

a) Reddy, G. Thippa, M. Praveen Kumar Reddy, Kuruva Lakshmanna, Rajesh Kaluri, Dharmendra Singh Rajput, Gautam Srivastava, and Thar Baker. "Analysis of dimensionality reduction techniques on big data." IEEE Access 8 (2020): 54776-54788.

b)Rehman, Zaka Ur, M. Sultan Zia, Giridhar Reddy Bojja, Muhammad Yaqub, Feng Jinchao, and Kaleem Arshid. "Texture based localization of a brain tumor from MR-images by using a machine learning approach." Medical hypotheses 141 (2020): 109705.

Reviewer #4: 1. List out the main contributions of the proposed work.

2. The related works can be summarized as a table.

3. Some of the recent works such as the following can be discussed in the paper "An ensemble machine learning approach through effective feature extraction to classify fake news, Variance ranking attributes selection techniques for binary classification problem in imbalance data".

4. Compare the current work with recent state-of-the-art.

5. Present a detailed analysis on the results obtained.

6. The english language used in the paper can be improved.

6. PLOS authors have the option to publish the peer review history of their article (what does this mean?). If published, this will include your full peer review and any attached files.

Reviewer #1: **Yes: **Rutvij H Jhaveri

Reviewer #2: No

Reviewer #3: No

Reviewer #4: No

---

## [Author Response · Author response to Decision Letter 0]

23 Apr 2021

we have reviewed the manuscript based on your comments and suggestions

---

## [Decision Letter · Decision Letter 1]

6 May 2021

Designing a Hybrid Dimension Reduction for Improving the Performance of Amharic News Document Classification

PONE-D-21-09681R1

Dear Dr. Endalie,

We’re pleased to inform you that your manuscript has been judged scientifically suitable for publication and will be formally accepted for publication once it meets all outstanding technical requirements.

Kind regards,

Thippa Reddy Gadekallu

Academic Editor

PLOS ONE

Additional Editor Comments (optional):

Reviewers' comments:

Reviewer's Responses to Questions

**Comments to the Author**

1. If the authors have adequately addressed your comments raised in a previous round of review and you feel that this manuscript is now acceptable for publication, you may indicate that here to bypass the “Comments to the Author” section, enter your conflict of interest statement in the “Confidential to Editor” section, and submit your "Accept" recommendation.

Reviewer #2: (No Response)

Reviewer #4: All comments have been addressed

2. Is the manuscript technically sound, and do the data support the conclusions?

Reviewer #2: Yes

Reviewer #4: Yes

3. Has the statistical analysis been performed appropriately and rigorously? 

Reviewer #2: Yes

Reviewer #4: Yes

4. Have the authors made all data underlying the findings in their manuscript fully available?

Reviewer #2: Yes

Reviewer #4: Yes

5. Is the manuscript presented in an intelligible fashion and written in standard English?

Reviewer #2: Yes

Reviewer #4: Yes

6. Review Comments to the Author

Reviewer #2: The authors have not addressed my all comments, therefore, i recommend second round.

The authors have not addressed my all comments, therefore, i recommend second round.

Reviewer #4: The authors have addressed all the comments. Hence I recommend the paper to be accepted in the current form.

7. PLOS authors have the option to publish the peer review history of their article (what does this mean?). If published, this will include your full peer review and any attached files.

Reviewer #2: No

Reviewer #4: No

---

## [Editor Report · Acceptance letter]

12 May 2021

PONE-D-21-09681R1 

Designing a Hybrid Dimension Reduction for Improving the Performance of Amharic News Document Classification 

Dear Dr. Endalie:

I'm pleased to inform you that your manuscript has been deemed suitable for publication in PLOS ONE. Congratulations! Your manuscript is now with our production department. 

Kind regards, 

on behalf of

Dr. Thippa Reddy Gadekallu 

Academic Editor

PLOS ONE